# Genomic analysis of carbapenemase-encoding plasmids and antibiotic resistance in carbapenem-resistant *Klebsiella pneumoniae* isolates from Vietnam, 2021

Lisa Göpel,[1] Le Thi Kieu Linh,[2,3] Bui Tien Sy,[2,4] Sébastien Boutin,[1,5] Simone Weikert-Asbeck,[1] Elias Eger,[6] Susanne Hauswaldt,[1] Truong Nhat My,[2,4] Kaan Kocer,[1,5] Nguyen Trong The,[2,4] Jan Rupp,[1,5,7] Le Huu Song,[2,3,4] Katharina Schaufler,[6,8] Thirumalaisamy P. Velavan,[2,3,9,10] Dennis Nurjadi[1,2,5]

**ABSTRACT** Carbapenem resistance in gram-negative rods is increasing in low- and middle-income countries. We conducted a single-center study to identify carbapenemase-encoding plasmids in carbapenem-resistant *Klebsiella pneumoniae* isolates causing human infections in Vietnam. The secondary objective was to investigate the prevalence of multidrug-resistant (MDR) and hypervirulent *K. pneumoniae* in this setting. Our genomic analysis study characterized 105 of 245 clinical *K. pneumoniae* isolates at the 108 Military Hospital in Hanoi, Vietnam, collected from intensive care unit and regular wards between 1 January 2021 and 31 December 2021. All isolates were characterized using long- and short-read sequencing, followed by hybrid assembly. Comprehensive genomic analysis was performed to identify carbapenemase-encoding plasmids, complemented by extended antibiotic susceptibility testing for commonly used and novel antibiotics. We observed a high prevalence of NDM-4-related carbapenem resistance (30.5%, 32/105) mostly carried by a specific 83-kb IncFII plasmid co-carrying the $bla_{TEM-1}$ (46.9%, 15/32). The genomic content of the $bla_{NDM-4}$-harboring plasmids is highly variable. While $bla_{OXA-181}$ and $bla_{OXA-48}$ were predominantly located on an IncX3 and an IncL plasmid, respectively, the majority of plasmids harboring $bla_{KPC-2}$ were not related to any named Inc-type. All isolates exhibited the MDR phenotype; however, the majority remained susceptible to the siderophore-cephalosporin cefiderocol (79%, 83/105). All isolates were susceptible to aztreonam/avibactam. In addition, we identified a hypervirulent, carbapenem-resistant *K. pneumoniae* ST23 strain, confirmed through both *in vitro* and *in vivo* experiments. Our study provides insights into plasmids harboring the carbapenemases New Delhi metallo-β-lactamase, oxacillinase-48 like, and *K. pneumoniae* carbapenemase-2 circulating in Vietnam.

**IMPORTANCE** Carbapenem resistance in *Klebsiella pneumoniae* is a major public health threat, especially in low- and middle-income countries. This study examined resistant strains from a hospital in Vietnam to understand how they spread and which antibiotics might still work. We found that a significant number of these bacteria carried resistance genes on different types of plasmids. Despite their resistance to many antibiotics, most strains remained susceptible to newer substances like cefiderocol and aztreonam/avibactam. Alarmingly, we also identified a hypervirulent strain that is carbapenem resistant, potentially posing an even greater risk to patients. This research provides insight into the epidemiology of the carbapenemase gene-harboring plasmids in a Vietnamese hospital. Understanding these resistance patterns can help guide antibiotic use and policy decisions to combat the growing threat of multidrug-resistant infections in Vietnam.

Address correspondence to Dennis Nurjadi, dennis.nurjadi@uni-luebeck.de.

Lisa Göpel, Le Thi Kieu Linh, and Bui Tien Sy contributed equally to this article. The order of the shared first author was determined by increasing order of the surnames.

Thirumalaisamy P. Velavan and Dennis Nurjadi contributed equally to this article.

D.N. received speaker's honoraria from Shionogi and Cepheid outside the scope of this work. All other authors have no conflicts of interest.

See the funding table on p. 12.

**KEYWORDS** NDM-4, KPC-2, OXA-48, carbapenemase-producing *Klebsiella pneumoniae*, hypervirulent *Klebsiella pneumoniae*, Vietnam

*K*lebsiella pneumoniae, as a member of the Enterobacterales, is considered a commensal of the gut and can colonize healthy individuals without causing infection (1). However, under certain circumstances, *K. pneumoniae* can transition from an asymptomatic colonizer to a pathogen leading to infections, such as pneumonia, urinary tract infections, bloodstream infections, and wound infections (2). In recent years, *K. pneumoniae* infections have become a major clinical challenge due to the ability of *K. pneumoniae* to develop or acquire resistance to various antimicrobial agents, leading to the emergence of multidrug-resistant (MDR) phenotypes (3).

The emergence of antimicrobial resistance in *K. pneumoniae* has been attributed to multiple factors, including antibiotic overuse in both clinical and agricultural sectors. The global success of *K. pneumoniae* can generally be traced back to specific clonal lineages, sometimes associated with multidrug resistance and the convergence of (hyper-)virulence and resistance (4–7). In Asia, the emergence of carbapenem-resistant *K. pneumoniae* has poses a significant challenge for public health and clinical practice (6, 8). Carbapenem resistance in *K. pneumoniae* is often conferred by the production of carbapenemases, such as *Klebsiella pneumoniae* carbapenemase-2 (KPC-2), New Delhi metallo-β-lactamase (NDM), or oxacillinase (OXA)-48 like. The localization of carbapenemase genes on various plasmids and their spread through horizontal gene transfer in hospitals has been reported, with plasmid-mediated transmission accounting for 50% of carbapenemase-producing Enterobacterales (9). A recent study by Pham et al. revealed the presence of $bla_{KPC-2}$ (57.1%, 184/322) and $bla_{NDM}$ (50%, 179/322) genes in meropenem-resistant *K. pneumoniae* strains isolated in 2017 and 2018 from patient and environmental samples in the intensive care units (ICUs) of two hospitals in Vietnam (8). The authors identified a putative self-transmissible IncN plasmid harboring $bla_{KPC-2}$ in the majority of $bla_{KPC-2}$-positive isolates (97.8%, 180/184). We recently reported the co-resistance of carbapenem-resistant *K. pneumoniae* isolates to colistin (37%, 39/105) (10). These isolates were collected in 2021 from the ICU and regular wards of the 108 Military Hospital in Hanoi, Vietnam. $bla_{NDM}$ was the most prevalent carbapenemase (48/103, 46.6%), followed by $bla_{OXA-48}$ like (43/103, 41.7%) and $bla_{KPC-2}$ (43/103, 41.7%). Although 30.1% (31/103) of the carbapenemase producers carried two different carbapenemase-encoding genes and most carbapenemases were identified on plasmids, no additional analysis of carbapenemase-encoding plasmids was performed. Data on the localization of the carbapenemase genes $bla_{KPC-2}$, $bla_{NDM}$, and $bla_{OXA-48}$ like on plasmids in *K. pneumoniae* isolates from hospital patients in Vietnam and the characterization of these plasmids are scarce.

With limited therapeutic options for infections caused by carbapenem-resistant *K. pneumoniae*, novel treatment strategies are urgently needed. In particular, the treatment of metallo-β-lactamase (MBL) producers, such as NDM, is especially challenging since this potent β-lactamase is able to degrade almost all β-lactam drugs, and there are no effective β-lactamase inhibitors, which can reliably inhibit this enzyme approved for clinical use. Several novel substances or combinations of drugs have been identified as showing great potential, including the siderophore-conjugated cephalosporin, cefiderocol, and the combination of β-lactams and β-lactamase inhibitors, such as aztreonam/avibactam (11). However, considering the local molecular epidemiology of carbapenem resistance, data on the efficacy of these substances are important in clinical practice.

The objective of the present study was to investigate and characterize carbapenemase-encoding plasmids in carbapenemase-resistant *K. pneumoniae* isolates from a previously described collection of 105 strains obtained during routine microbiological diagnostics in a tertiary hospital in Hanoi, Vietnam. As a secondary objective, this study aimed to investigate the prevalence of MDR and hypervirulent *K. pneumoniae* in this setting and their susceptibility to novel antibiotics.

## MATERIALS AND METHODS

### Study setting and study population

This study is based on a collection of 105 carbapenem-resistant *K. pneumoniae* isolates that we have previously described (10). Briefly, the representative samples used were collected from hospitalized patients from 22 clinical departments at the 108 Military Central Hospital in Hanoi, Vietnam, including ICU and normal wards, between 1 January 2021 and 31 December 2021. Of the 524 *K. pneumoniae* isolates from clinical samples obtained from various clinical specimens (respiratory, blood, urine, and body fluid samples) detected in the routine microbiological diagnostic in 2021, 245 (47%) were carbapenem-resistant (MIC imipenem >4 mg/L and/or MIC meropenem >8 mg/L). We performed extended susceptibility testing and whole-genome sequencing (WGS) on 105 (43%) non-duplicate strains from 101 patients (4 patients have 2 morphologically different strains). The definition of non-duplicate strains isolated from a single patient is either morphologically different or diverging antibiotic susceptibility profiles.

### Bacterial identification and antibiotic susceptibility testing

The bacterial culture was carried out in accordance with standard procedures meeting ISO 15189:2012 standards in the routine microbiological diagnostic laboratory. Species identification was performed by matrix-assisted laser desorption ionization time-of-flight mass spectrometry [matrix-assisted laser desorption ionization time-of-flight (MALDI-TOF), Vitek MS, BioMérieux]. The initial antibiotic susceptibility testing (AST) was conducted as part of the routine microbiological diagnostic procedures in Vietnam. This testing was performed using the Vitek2 Compact (BioMérieux) automated AST system, on the AST-N204 card. Furthermore, AST was performed by broth micro-dilution (BMD; Sensititre EUMDRXXF, Thermo Fisher, Germany). The MICs of cefidero-col were determined by utilizing the broth microdilution method with iron-depleted cation-adjusted Mueller-Hinton broth (CA-MHB) (12), while aztreonam/avibactam MICs were determined by MIC gradient test (Etest) (bestbion dx GmbH, Germany). For cefiderocol-resistant isolates, BMD was performed using iron-depleted CA-MHB, with the addition of 100 mg/L dipicolinic acid (DPA). This chemical compound is capable of chelating metal ions, thereby inhibiting the activity of MBL, and is useful to determine whether the cefiderocol resistance was due to overexpression or hydrolysis activity of the NDM enzyme. Antibiotic susceptibility was interpreted according to the EUCAST clinical breakpoints v13. *Escherichia coli* ATCC25922 was used as a quality control strain.

### Whole-genome sequencing and bioinformatic analysis

Whole-genome sequencing and bioinformatic analysis were performed as previously described (10). The assembled genomes were screened for plasmid and typed using the MOB-suite (command mob_typer with default parameter) (13, 14). Only the plasmid with complete sequence and typed as plasmid by the MOB-suite were used for the subsequent analysis. Complete plasmids were compared using average nucleotide identity (ANI) with ANIclustermap (v1.1.0). The tree was used for the visualization of the plasmid primary and secondary clusters from the MOB-suite, and clusters based on ANI (ANI >99.9%) were visualized using Clinker (15) and gggenome R package. In order to identify any mutations that could potentially contribute to cefiderocol resistance despite the addition of DPA, a comparative analysis between resistant isolates and all susceptible isolates (including the resistant isolates, which become sensitive with the addition of DPA) was conducted. Initial mapping was performed against the corresponding wild-type reference strain KP21315 (GenBank accession number: CP124702) to facilitate the annotation of the single nucleotide polymorphisms (SNPs), and SNP calling was performed using snippy (https://github.com/tseemann/snippy). All the SNPs found in the sensitive strains or in the resistant isolates, which become sensitive with the addition of DPA, were considered as polymorphisms non-related to cefiderocol resistance.

## Phenotypic confirmation of hypervirulence

To evaluate the hypervirulent phenotype, Kp084 was challenged in a variety of *in vitro* and *in vivo* virulence assays, including determining hypermucoviscosity, siderophore secretion, resistance to complement-mediated killing, survival in bile salts, and the ability to cause mortality in larvae of the greater wax moth larvae (*Galleria mellonella*). All phenotypic assays were based on previously published protocols (16, 17), with minor modifications as described below. For reference, we used a well-characterized hypervirulent *K. pneumoniae* ST420 (PBIO2030 [16]) and *E. coli* K-12 W3110. Further details are included in the Supplementary Material.

## Data analysis

Descriptive statistical analysis was performed using STATA/BE v18.0 (StataCorp, USA). Statistics for the validation of the hypervirulence phenotype were performed using GraphPad Prism 8.0 (https://www.graphpad.com/). The non-parametric Kruskal-Wallis test was applied to compare bacterial groups using median values. A *P*-value of <0.05 was considered statistically significant.

## RESULTS

In 2021, a total of 524 non-duplicate *K. pneumoniae* isolates were collected from various clinical specimens in the routine microbiological diagnostic, of which 245 (47%) were phenotypically resistant to meropenem and/or imipenem. Of the 245 carbapenem-resistant isolates, every second isolate was selected (122, 50%) for WGS. However, 17 isolates could not be recovered, and only 105 (43%) isolates from 101 patients (4 patients had 2 morphologically distinct isolates) underwent WGS. Information regarding quality control of the sequencing of 105 isolates, as well as antimicrobial resistance genes, virulence genes, and carbapenemase-encoding plasmids, can be found in the Supplementary Data Set and was already reported in part recently (10).

## Carbapenemase-encoding plasmids

A total of 79.1% (106/134) of carbapenemase genes were identified on plasmids in 86 of the 105 analyzed isolates. The plasmids carrying $bla_{NDM-4}$ could be clustered based on average nucleotide identity into 14 variants. The most abundant variant (A: 15/29) was an 83-kb IncFII plasmid, also carrying $ble_{MBL}$, $bla_{TEM-1}$, and $rmtB1$ genes, which were all identified in isolates belonging to ST16 (Fig. 1). The $bla_{NDM-4}$-carrying plasmid positive for replicon-type repB(R1701) was identified in 11 isolates from 7 distinct multi-locus sequence types (ST15, ST273, ST307, ST438, ST580-1LV, ST656, and ST789), while only 1 isolate was found to harbor $bla_{NDM-4}$ on an IncFIB plasmid (ST438). Further analysis revealed identical MOB-suite primary and secondary clusters for 15/16 IncFII plasmids and 9/11 repB(R1701) plasmids (Supplementary Data Set; Fig. S1 and S2). Plasmids that are assigned to the same secondary MOB cluster may be regarded as sufficiently related to be potentially part of an outbreak.

The plasmids carrying $bla_{NDM-1}$ were less diverse with only four variants. The most prevalent variant (variant A, 4/9) was an 86-kb IncFII plasmid co-carrying $bla_{CTX-M-15}$, $bla_{TEM-1}$ (two copies), $fosA3$, $rmtB1$, and $ble_{MBL}$ (Fig. 2). MOB-suite analysis revealed the same primary (AA327) and secondary cluster (AI227) for all IncFII plasmids (7/9), which were identified in seven isolates belonging to ST16 (Supplementary Data Set; Fig. S1 and S3).

All 16 IncX3 plasmids carrying $bla_{OXA-181}$ exhibited a high degree of conservation across the entire population (ANI: 99.8%–100%, identical MOB-suite primary and secondary cluster) and were found to carry the $qnrS1$ gene. These plasmids were identified in isolates belonging to ST16 (15 isolates) and ST23 (1 isolate). The same high level of similarity was observed in 15/16 $bla_{OXA-48}$-carrying plasmids, which all harbored only $bla_{OXA-48}$ and belonged to the IncL type (Supplementary Data Set; Fig. S4 and S5).

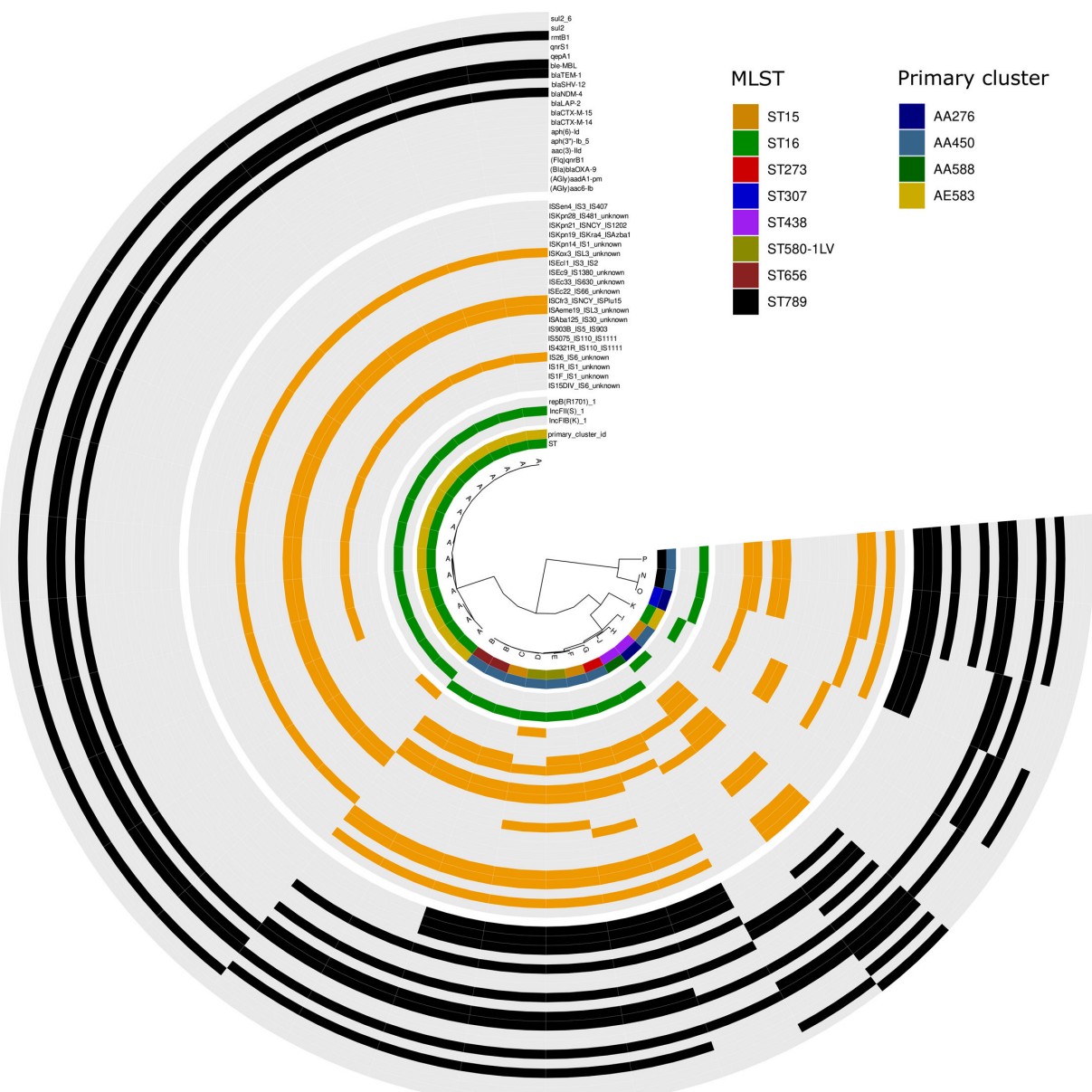

**FIG 1** ANI-based dendrogram and genomic content of the *bla*<sub>NDM-4</sub>-carrying plasmids. Complete plasmids (*n* = 29) were compared using average nucleotide identity with ANIclustermap (v.1.1.0) and plasmid cluster. Four sections of the circle (separated by white gaps) represent, from inner to outer: (i) ST and primary cluster-ID color code as in the legend, (ii) replicon types (presence of the Inc-type in green, absence in gray), (iii) insertion sequence elements (presence of the IS in orange, absence in gray), and (iv) antimicrobial resistance genes present on the plasmids (presence of the AMR gene in black, absence in gray).

One isolate (ST16) was found to carry a plasmid containing two replicon types (IncL and IncR) and multiple antimicrobial resistance genes (Fig. 3).

A total of 36 *bla*<sub>KPC-2</sub>-carrying plasmids were identified and categorized into 14 variants (ANI) and 5 different primary clusters (MOB-suite). The majority of plasmids (22/36) carrying *bla*<sub>KPC-2</sub> were not related to any named Inc-type and carry *qnrS1*, *bla*<sub>TEM-1</sub>, *Mrx*, and *mph*(A) in addition to *bla*<sub>KPC-2</sub>. These were identified in ST16 (16 isolates) and ST11 (6 isolates) (Fig. 4). Furthermore, seven *bla*<sub>KPC-2</sub>-carrying IncN plasmids were detected and primarily found in isolates belonging to ST15 (five isolates). The remaining plasmids carried the repB(R1701) replicon type and were identified in ST16 (six isolates) and ST16-1LV (one isolate). While the primary and secondary MOB clusters were found to be identical for IncN and repB(R1701), respectively, the analysis of ANI

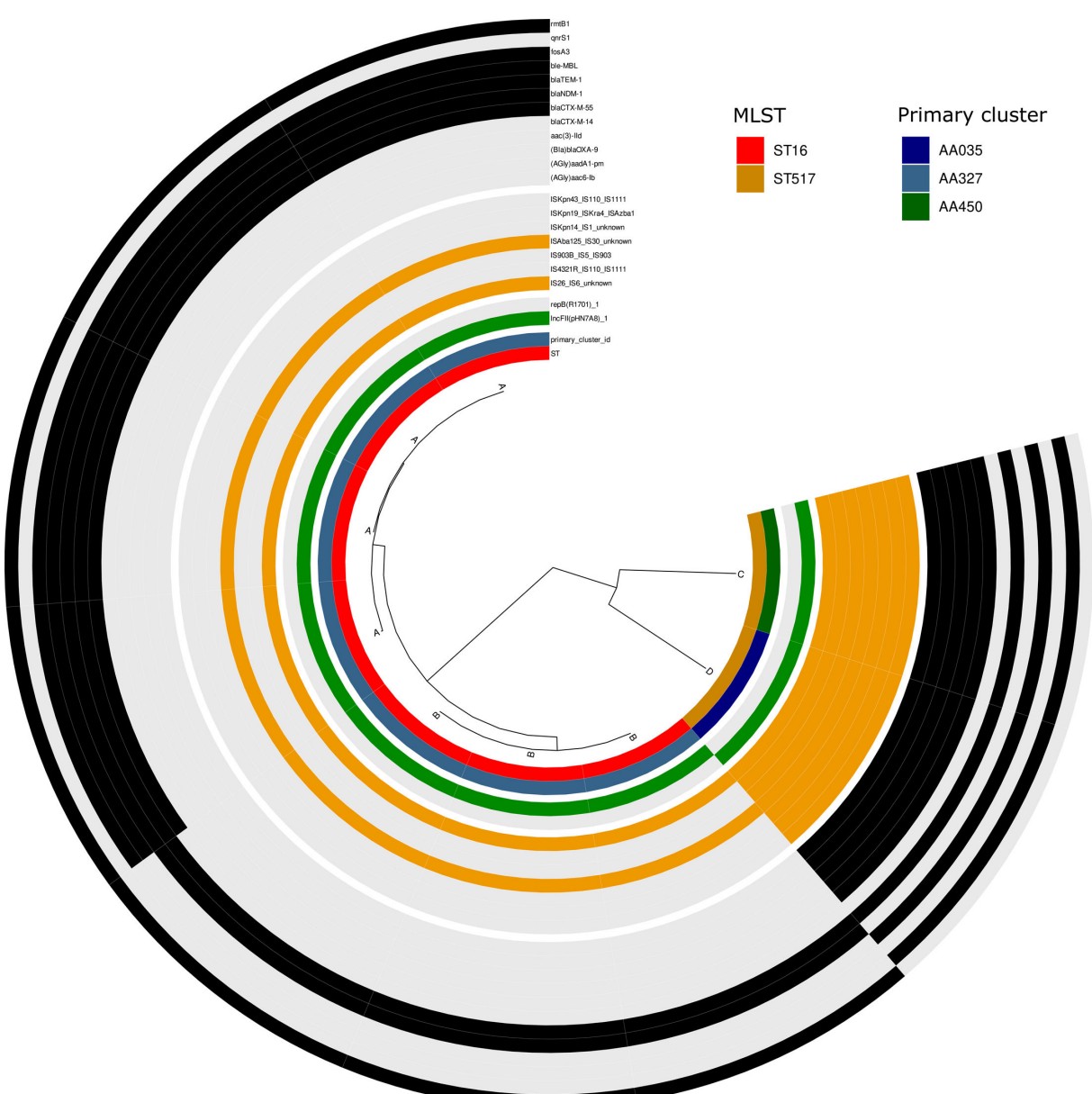

**FIG 2** ANI-based dendrogram and genomic content of the $bla_{NDM-1}$-carrying plasmids. Complete plasmids ($n$ = 9) were compared using average nucleotide identity with ANIclustermap (v.1.1.0) and plasmid cluster. The four sections of the circle (separated by white gaps) represent, from inner to outer: (i) ST and primary cluster-ID color code as in the legend, (ii) replicon types (presence of the Inc-type in green, absence in gray), (iii) insertion sequence elements (presence of the IS in orange, absence in gray), and (iv) antimicrobial resistance genes present on the plasmids (presence of the AMR gene in black, absence in gray).MLST, multi-locus sequence type.

revealed clustering in three variants for both plasmid types (Supplementary Data Set; Fig. S4 and S6).

A total of 20 carbapenemase-producing *K. pneumoniae* isolates (all belonging to ST16) carrying 2 different carbapenemase-encoding plasmids were identified. The $bla_{NDM-4}$-encoding IncFII plasmid (cluster A) and the IncX3-harboring $bla_{OXA-181}$ plasmid were present in 14 isolates. Six isolates harbored different plasmid cluster combinations of $bla_{KPC-2}$-carrying plasmids and $bla_{NDM-1}$-encoding plasmids IncFII (Supplementary Data Set). Further visualization and comparison of the overall plasmid structures of carbapenemase-encoding plasmids from this study can be found in the Supplementary Material.

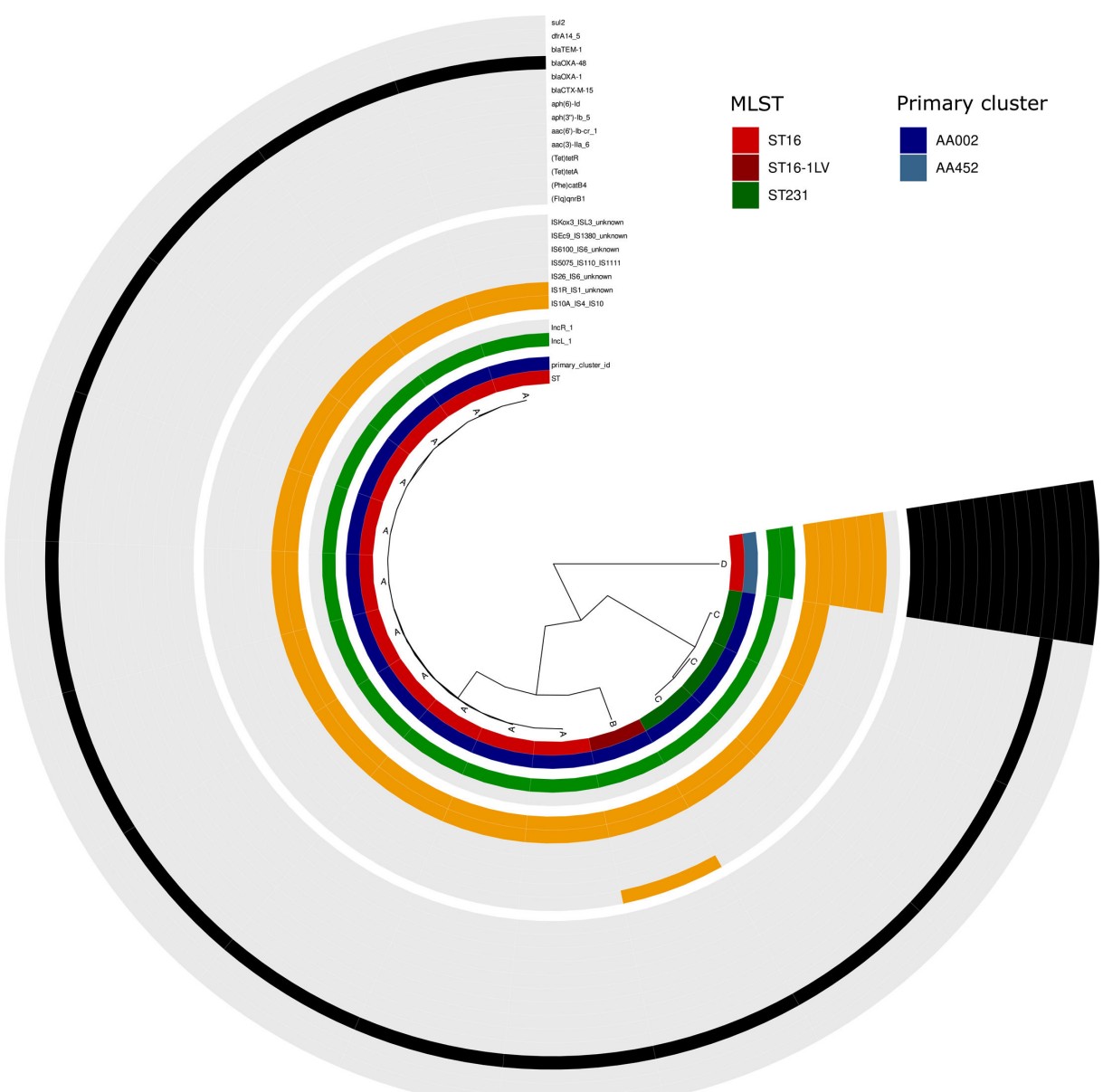

**FIG 3** ANI-based dendrogram and genomic content of the $bla_{OXA-48}$-carrying plasmids. Complete plasmids ($n = 16$) were compared using average nucleotide identity with ANIclustermap (v.1.1.0) and plasmid cluster. The four sections of the circle (separated by white gaps) represent, from inner to outer: (i) ST and primary cluster-ID color code as in the legend, (ii) replicon types (presence of the Inc-type in green, absence in gray), (iii) insertion sequence elements (presence of the IS in orange, absence in gray), and (iv) antimicrobial resistance genes present on the plasmids (presence of the AMR gene in black, absence in gray).

## Phenotypic resistance to β-lactam antibiotics

All isolates resistant to three or more antimicrobial substance classes were considered to be MDR. The majority of isolates exhibited resistance to aztreonam and cefepime (98.1% and 96.2%, respectively), while resistance rates for imipenem and meropenem ranged from 81% to 85.7%, respectively. Two isolates (Kp052 and Kp072) showed susceptibility to meropenem despite the presence of carbapenemase-encoding genes $bla_{NDM-4}$ and $bla_{OXA-48}$, respectively. Seven of the nine imipenem-susceptible isolates harbored at least one carbapenemase gene, specifically $bla_{OXA-48}$ (three isolates), $bla_{KPC-2}$ (two isolates), $bla_{NDM-4}$ (one isolate), and a combination of $bla_{KPC-2}$ and $bla_{NDM-1}$ (one isolate). The AST results also revealed high resistance rates for piperacillin/tazobactam (105/105, 100%) and ceftolozane/tazobactam (101/105, 96.2%), while the combination

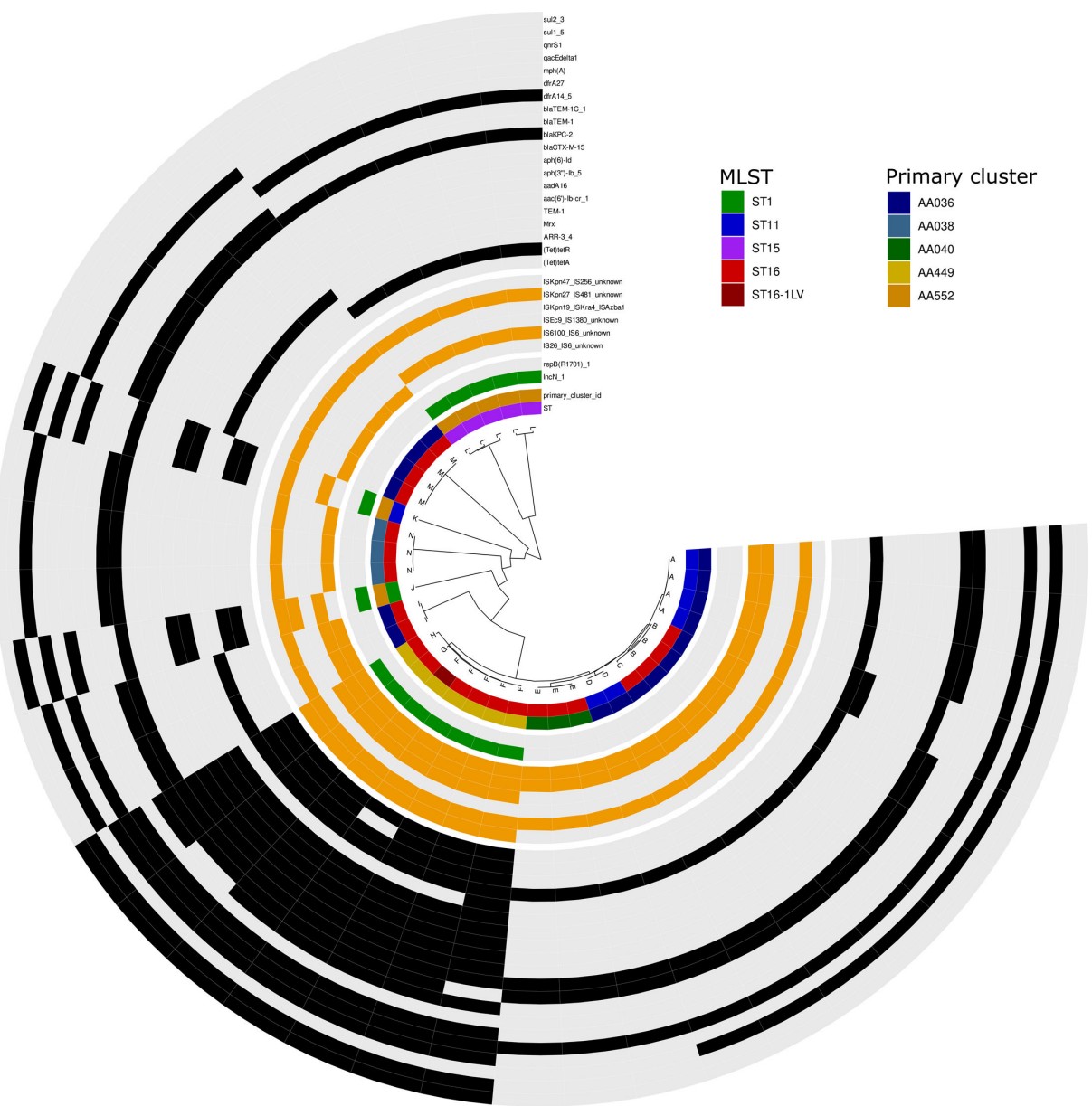

**FIG 4** ANI-based dendrogram and genomic content of the $bla_{KPC-2}$-carrying plasmids. Complete plasmids ($n$ = 36) were compared using average nucleotide identity with ANIclustermap (v.1.1.0) and plasmid cluster. The four sections of the circle (separated by white gaps) represent, from inner to outer: (i) ST and primary cluster-ID color code as in the legend, (ii) replicon types (presence of the Inc-type in green, absence in gray), (iii) insertion sequence elements (presence of the IS in orange, absence in gray), and (iv) antimicrobial resistance genes present on the plasmids (presence of the AMR gene in black, absence in gray). MLST, multi-locus sequence type.

of ceftazidime/avibactam exhibited a 44.8% (47/105) resistance rate among isolates. The combination of carbapenem antibiotics with β-lactamase inhibitors demonstrated a lower resistance rate for imipenem/relebactam (63/105, 60%) and meropenem/vabor-bactam (61/105, 58.1%) compared to single imipenem and meropenem exposure. All isolates were susceptible to aztreonam in combination with avibactam (MIC ≤4 mg/L). Concerning non-β-lactam antibiotics, the resistance rate to tobramycin was high (91/105, 86.7%) but lower for amikacin (54/105, 51.4%), eravacycline (54/105, 51.4%), and tigecycline (51/105, 48.6%) (Supplementary Data Set).

Twenty-two of 105 (21%) tested isolates were resistant to cefiderocol (MIC ≥4 mg/L). Of these, 10 isolates were positive for KPC-2, and 9 isolates were positive for NDM. When

adding 100 µg/mL of MBL inhibitor, DPA, into the iron-depleted CA-MHB, 15/22 (68.2%) of the initially cefiderocol-resistant isolates became susceptible to cefiderocol (including all nine NDM-producing isolates). For the remaining seven cefiderocol-resistant isolates (all KPC-2-producing *K. pneumoniae*), specific mutations observed only in these strains were detected in the *ftsI* gene, which encodes for the penicillin-binding protein 3 and is one of the known target proteins of cefiderocol (five of seven isolates) as well as genes coding for iron binding and transport (*fhuA*, *fepA*, *fepD,* and *nqrF*; six of seven isolates). Furthermore, an SNP in the diacylglycerol kinase, which is associated with metal binding, was identified in one isolate (Supplementary Data Set).

## Hypervirulent ST23 *K. pneumoniae*

In one isolate of *K. pneumoniae* ST23 (Kp084), obtained from the tracheal aspirate of a neurosurgical patient experiencing post-operative respiratory infection and carrying both $bla_{KPC2}$ and $bla_{OXA-48}$, biomarkers linked to hypervirulent *K. pneumoniae* were detected. The yersiniabactin operon and the colibactin operon were chromosomally localized. The colibactin operon was inserted via a transposable element (insertion sequence IS102, (IS5 family, group IS903) upstream of the yersiniabactin locus. The salmochelin operon and the aerobactin operon were located on a 223-kb plasmid carrying two replicon types IncHI1B(pNDM-MAR)/repB and several cation efflux operons

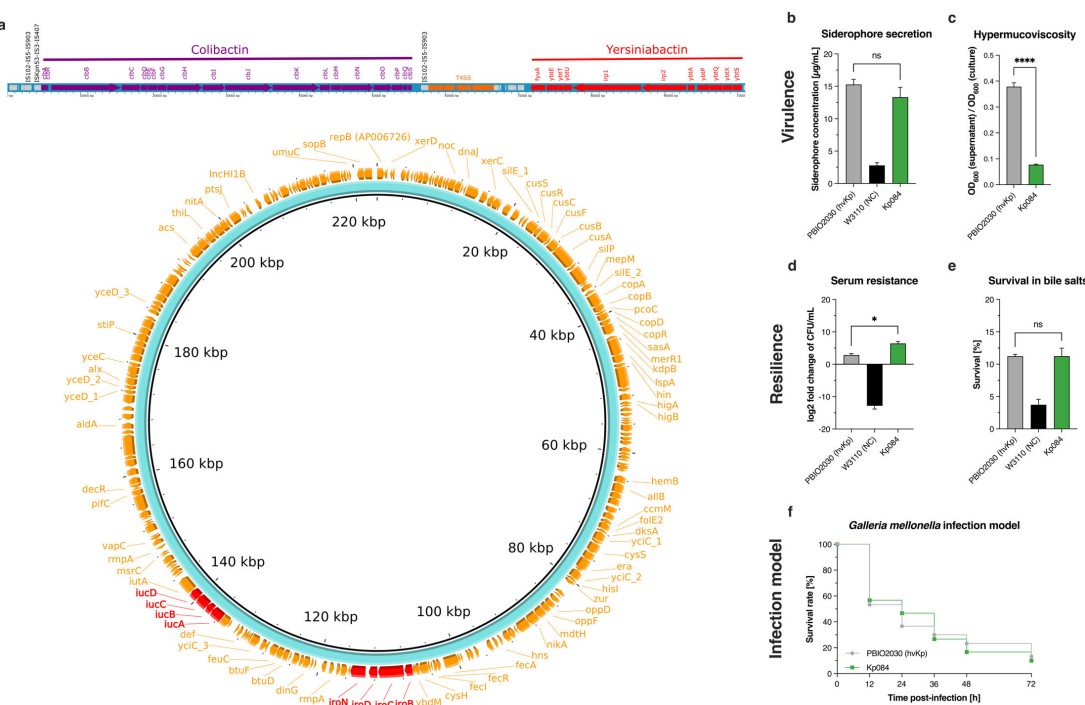

FIG 5   Loci of virulence determinant conferring the hypervirulence phenotype in isolate Kp084. (a) The gene clusters conferring colibactin and yersiniabactin production are located within the chromosome. The colibactin operon was inserted by an IS5 family transposable element (IS102) upstream of the yersiniabactin operon. The aerobactin operon (*iucABCD iutA* genes) and the salmochelin operon (*iroBCDN* genes), indicated by red font, are located on an ~220-kb IncHI1b plasmid. There were no resistance genes detected on this plasmid. (b) The amount of siderophore secreted is expressed as the mean of the siderophore concentration in the supernatant of the bacterial culture and the standard error (*n* = 4). (c) Determination of hypermucoviscosity by sedimentation assay (*n* = 3). Results are expressed as the mean ratio of the $OD_{600}$ of the supernatant after centrifugation at 1,000 × *g* for 5 min to the total $OD_{600}$ and the standard error. (d) Survival in 50% human serum (*n* = 3). Results are expressed as mean and standard error of log2 fold change in CFU/mL after 4 h of incubation in the presence of human serum. (e) Resilience against 50 mg/mL bile salts (*n* = 3). Results are shown as mean percent survival rates and standard error. (f) Kaplan-Meier plot of mortality in the *G. mellonella* larvae infection model (*n* = 3). Results are expressed as mean percent mortality after the injection of $10^5$ CFU per larva. The well-characterized PBIO2030 and the *E. coli* K12 W3110 were used as hypervirulent *K. pneumoniae* reference and negative control (NC), respectively. For all results, Kp084 was statistically compared to PBIO2030 using the Kruskal-Wallis test, and the following indicates the level of significance (*P* value): ns, not significant (*P* ≥ 0.05); *, *P* < 0.05; ****, *P* < 0.0001.

(*cus* and *fec*) or metal resistance gene (Fig. 5a). However, Kp084 did not exhibit a hypermucoid phenotype despite encoding hypermucoviscosity-modulating genes (e.g., *rmpA* and *rmpA2*). The hypervirulent trait of Kp084 was assessed using *in vitro* siderophore assays and *in vivo* G. *mellonella* survival assays with results comparable to the reference hypervirulent *K. pneumoniae* PBIO2030 (Fig. 5b through f), a previously characterized strain belonging to ST420 (17).

## DISCUSSION

The majority of carbapenemase genes identified were located on plasmids (106/134), which were predominantly of the IncX3 type for $bla_{OXA-181}$ and the IncL type for $bla_{OXA-48}$. Plasmids carrying $bla_{NDM-4}$ and $bla_{NDM-1}$ exhibited a higher degree of diversity, with the majority of these plasmids belonging to the IncFII type. These carbapenemase-encoding plasmid types have been identified in *Enterobacteriaceae* from multiple countries across several continents (18, 19). In contrast, the majority of plasmids harboring $bla_{KPC-2}$ were not related to any named Inc-type. In the present study, only 19.4% (7/36) $bla_{KPC-2}$ genes were identified on an IncN plasmid. Previous data from *K. pneumoniae* isolates collected in 2017 and 2018 from two hospitals in Vietnam demonstrated a much higher detection rate of $bla_{KPC-2}$ on one IncN plasmid, with a percentage of 97.8% (180/184) (8). The authors also reported the high prevalence of co-carriage of $bla_{KPC-2}$ and $bla_{NDM-1}$ in 164 out of 357 isolates. However, no plasmids were identified as harboring $bla_{NDM-1}$. Here, we report on the co-carriage of one $bla_{NDM-4}$-encoding IncFII plasmid and one IncX3 plasmid harboring $bla_{OXA-181}$ in 14 isolates, while 6 isolates carried both $bla_{KPC-2}$ and $bla_{NDM-1}$ on different plasmids. Although the clinical implications and significance of dual carbapenemase producers remain unclear, the presence of two β-lactamases, belonging to different classes, e.g., metallo- and serine β-lactamase, may further limit the clinical use of novel antibiotics by rendering them ineffective. Therefore, the increasing presence of dual carbapenemase producers and the localization of carbapenemase genes on plasmids should be closely monitored.

Due to the high prevalence of NDM producers in our study isolates, resistance to newer β-lactam and β-lactam/β-lactamase inhibitors was expected. Although cefiderocol is not yet used in Vietnamese hospitals, 21% of the carbapenem-resistant *K. pneumoniae* showed reduced susceptibility (increased MIC) to cefiderocol. A possible explanation for this phenomenon is the low hydrolytic activity of NDM toward cefiderocol or overexpression of NDM (20, 21). Indeed, the addition of DPA to inhibit NDM activity resulted in the reduction of the cefiderocol MIC to susceptible levels in all nine of the cefiderocol-resistant isolates, indicating that the MIC increase was mediated by the NDM activity. In those isolates that remained cefiderocol resistant despite the addition of DPA, several mutations were detected in genes associated with iron binding and transport. Resistance to cefiderocol in Enterobacterales has been attributed to mutations in the TonB-dependent siderophore receptors (22, 23). On a positive note, there was no resistance to the combination of aztreonam/avibactam in this study cohort. Although avibactam cannot withstand hydrolysis by NDM, the combination partner aztreonam is considered to be stable to NDM but not to other β-lactamases, and the combination with avibactam is thought to avoid degradation of aztreonam by other enzymes, such as AmpC and CTX-M (24).

We detected a hypervirulent and carbapenem-resistant *K. pneumoniae* ST23 (convergent pathotype). Although the *rmpA* and *rmpA2* genes associated with hypermucoviscosity (25) were present, our isolate did not exhibit a hypermucoid phenotype. Since we only detected a single case of hypervirulent *K. pneumoniae* in this study, we could not elucidate whether the lack of hypermucoviscosity is a natural evolutionary trajectory of this particular clone or a one-off deviation. The loss of hypermucoviscosity may pose a challenge for the identification of the hypervirulent phenotype in routine microbiological diagnostics as the hypervirulent phenotype is commonly associated with increased capsule production. In contrast to our study, previous studies from Vietnam

found no hypervirulent *K. pneumoniae* (8, 26). However, in these studies, the isolates were collected before 2018, while our *K. pneumoniae* isolates were collected between January and December 2021, suggesting that the hypervirulent *K. pneumoniae* ST23 may be emerging in Vietnam. Convergent pathotypes, such as *K. pneumoniae* ST307, have been described to cause outbreaks, and the spread of this clone should be closely monitored (4).

This study has several limitations. First, the present study was conducted exclusively with the clinical isolates from hospitalized patients from a single center. A total of 245 carbapenem-resistant *K. pneumoniae* were collected during the designated sampling period, of which a random selection of 105 isolates was subjected to further investigation via whole-genome sequence analysis. The strength of our study was the application of hybrid (long- and short-read) sequencing on all isolates, coupled with comprehensive phenotypic resistance testing. Furthermore, we found evidence for the occurrence of convergent pathotypes of *K. pneumoniae* that simultaneously combine resistance and enhanced virulence. This finding was validated by phenotypic testing and comparison with a well-characterized reference strain. However, we did not perform the mouse infection model, which is considered to be a more accurate *in vivo* assay for assessing hypervirulence compared to the *Galleria mellonella* assay (27). Despite this, we successfully demonstrated that the isolate identified in this study possesses the genomic characteristics associated with a hypervirulent phenotype.

In conclusion, our data demonstrate the localization of carbapenemase genes on different plasmids in carbapenemase-producing *K. pneumoniae* in Vietnam. Comprehensive antimicrobial susceptibility testing revealed high rates of resistance to the majority of the antibiotics tested. However, most isolates were susceptible to cefiderocol, and all isolates were susceptible to aztreonam-avibactam, suggesting that these drugs are promising for treating infections caused by MDR and extensively resistant Enterobacterales. Furthermore, the detection of MDR and hypervirulent *K. pneumoniae* ST23 in the hospital setting is alarming and should be closely monitored. Our data also highlight the importance of incorporating WGS to understand the transmission dynamics of MDR bacteria in Vietnam and to develop tailored and effective infection prevention and control measures.

## ACKNOWLEDGMENTS

K.S. received a grant from the Federal Ministry of Education and Research (BMBF, Germany) entitled "Disarming pathogens as a different strategy to fight antimicrobial-resistant gram negatives" (01KI2015). T.P.V. received PAN-ASEAN Coalition for Epidemic and Outbreak Preparedness (PACE-UP; DAAD Project ID: 57592343) and staff support through JPIAMR I-CRECT (BMBF-01KI2207).

The study was funded through grants from the PAN-ASEAN Coalition for Epidemic and Outbreak Preparedness (PACE-UP; DAAD Project ID: 57592343) and staff support through JPIAMR I-CRECT.

D.N. has received speaker's honoraria from Shionogi and Cepheid outside the scope of this work. All other authors reported no conflict of interest.

## AUTHOR AFFILIATIONS

[1]Institute of Medical Microbiology, University of Lübeck and University Hospital Schleswig-Holstein Campus Lübeck, Lübeck, Schleswig-Holstein, Germany

[2]Vietnamese - German Centre for Medical Research (VG-CARE), Hanoi, Vietnam

[3]Institute of Tropical Medicine, University of Tübingen, Tübingen, Germany

[4]108 Military Central Hospital, Hanoi, Vietnam

[5]German Center for Infection Research (DZIF), Partner Site Hamburg-Lübeck-Borstel-Riems, Hamburg-Lübeck-Borstel-Riems, Lübeck, Germany

[6]Epidemiology and Ecology of Antimicrobial Resistance (GEAR), Helmholtz Institute for One Health (HIOH), Helmholtz Centre for Infection Research (HZI), Greifswald, Germany

[7]Infectious Disease Clinic, University of Lübeck and University Hospital Schleswig-Holstein Campus Lübeck, Lübeck, Germany

[8]University Medicine Greifswald, Greifswald, Germany

[9]Faculty of Medicine, Duy Tan University, Da Nang, Vietnam

[10]German Center for Infection Research (DZIF), Partner Site Tübingen, Tübingen, Germany

## AUTHOR ORCIDs

Bui Tien Sy http://orcid.org/0000-0002-4615-0114

Elias Eger http://orcid.org/0000-0002-5514-8083

Truong Nhat My http://orcid.org/0000-0003-2436-7897

Kaan Kocer http://orcid.org/0000-0002-1134-4197

Jan Rupp http://orcid.org/0000-0001-8722-1233

Katharina Schaufler http://orcid.org/0000-0002-2669-8799

Thirumalaisamy P. Velavan http://orcid.org/0000-0002-9809-9883

Dennis Nurjadi http://orcid.org/0000-0002-1278-5939

## FUNDING

| Funder | Grant(s) | Author(s) |
| --- | --- | --- |
| Bundesministerium für Bildung und Forschung | 01KI2015 | Katharina Schaufler |
| Deutscher Akademischer Austauschdienst | 57592343 | Thirumalaisamy P. Velavan |
| Bundesministerium für Bildung und Forschung | BMBF-01KI2207 | Thirumalaisamy P. Velavan |

## AUTHOR CONTRIBUTIONS

Lisa Göpel, Data curation, Formal analysis, Methodology, Writing – original draft, Writing – review and editing | Le Thi Kieu Linh, Data curation, Formal analysis, Investigation, Methodology, Writing – original draft, Writing – review and editing | Bui Tien Sy, Data curation, Investigation, Methodology, Supervision, Writing – original draft, Writing – review and editing | Sébastien Boutin, Conceptualization, Data curation, Formal analysis, Methodology, Visualization, Writing – original draft, Writing – review and editing | Simone Weikert-Asbeck, Data curation, Investigation, Methodology, Writing – review and editing | Elias Eger, Data curation, Investigation, Methodology, Visualization, Writing – review and editing | Susanne Hauswaldt, Data curation, Investigation, Methodology, Writing – review and editing | Truong Nhat My, Data curation, Investigation, Methodology, Project administration, Writing – original draft, Writing – review and editing | Kaan Kocer, Formal analysis, Investigation, Methodology, Writing – original draft, Writing – review and editing | Nguyen Trong The, Data curation, Formal analysis, Investigation, Methodology, Writing – review and editing | Jan Rupp, Data curation, Formal analysis, Methodology, Resources, Writing – review and editing | Le Huu Song, Methodology, Resources, Supervision, Writing – review and editing | Katharina Schaufler, Data curation, Formal analysis, Funding acquisition, Investigation, Methodology, Visualization, Writing – original draft, Writing – review and editing | Thirumalaisamy P. Velavan, Data curation, Funding acquisition, Methodology, Resources, Supervision, Writing – original draft, Writing – review and editing | Dennis Nurjadi, Conceptualization, Formal analysis, Investigation, Methodology, Project administration, Resources, Supervision, Visualization, Writing – original draft, Writing – review and editing

## DATA AVAILABILITY

The draft genomes presented in this study can be found in the NCBI Genbank repositories under the Bioproject PRJNA1043438.

## ETHICS APPROVAL

All participants gave informed consent before taking part in the study. The study was approved by the Institutional Review Board of the 108 Military Central Hospital, Hanoi, Vietnam (108MCH/RES/MENTNGITIS-V-D3-25-04-2017) and (108MCH/RES/I-CRECT-V1.1-D2-06-04-2021).

## ADDITIONAL FILES

The following material is available online.

### Supplemental Material

**Supplemental data set (Spectrum03115-24-s0001.xlsx).** Overview of complete carbapenemase-encoding plasmids analyzed in this study.
**Supplemental material (Spectrum03115-24-s0002.pdf).** Supplemental Methods; Fig. S1 to S6.

### Open Peer Review

**PEER REVIEW HISTORY (review-history.pdf).** An accounting of the reviewer comments and feedback.

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
