## [Reviewer comments · Microbiology Spectrum]

Microbiology Spectrum

Genomic analysis of carbapenemase-encoding plasmids and antibiotic resistance in carbapenem-resistant *Klebsiella pneumoniae* isolates from Vietnam, 2021

Lisa Göpel, Le Thi Kieu Linh, Bui Tien Sy, Sébastien Boutin, Simone Weikert-Asbeck, Elias Eger, Susanne Hauswaldt, Truong Nhat My, Kaan Kocer, Nguyen Trong The, Jan Rupp, Le Huu Song, Katharina Schaufler, Thirumalaisamy Velavan, and Dennis Nurjadi

Corresponding Author(s): Dennis Nurjadi, Universitat zu Lubeck

Review Timeline:

Submission Date:	February 3, 2025
Editorial Decision:	February 20, 2025
Revision Received:	February 20, 2025
Accepted:	March 20, 2025

Editor: Ahmed Babiker

Reviewer(s): The reviewers have opted to remain anonymous.

Transaction Report:

DOI: <https://doi.org/10.1128/spectrum.03115-24>

Re: Spectrum03115-24 (Genomic analysis of carbapenemase-encoding plasmids and antibiotic resistance in carbapenem-resistant *Klebsiella pneumoniae* isolates from Vietnam, 2021)

Dear Prof. Dennis Nurjadi:

Thank you for the privilege of reviewing your work and addressing the prior reviewer comments. I am pleased to inform you that your manuscript has been editorially accepted. However, you must complete the submission form before acceptance.

Revision Guidelines

Sincerely,
Ahmed Babiker
Editor
Microbiology Spectrum

Re: Spectrum03115-24R1 (Genomic analysis of carbapenemase-encoding plasmids and antibiotic resistance in carbapenem-resistant *Klebsiella pneumoniae* isolates from Vietnam, 2021)

Dear Prof. Dennis Nurjadi:

Your manuscript has been accepted, and I am forwarding it to the ASM production staff for publication. Your paper will first be checked to make sure all elements meet the technical requirements. ASM staff will contact you if anything needs to be revised before copyediting and production can begin. Otherwise, you will be notified when your proofs are ready to be viewed.

Sincerely,
Ahmed Babiker
Editor
Microbiology Spectrum